# Pharmacologic IRE1/XBP1s activation promotes systemic adaptive remodeling in obesity

Aparajita Madhavan[1,4], Bernard P. Kok[1,4], Bibiana Rius[1], Julia M. D. Grandjean[1], Adekunle Alabi[1], Verena Albert[1], Ara Sukiasyan[1], Evan T. Powers[2], Andrea Galmozzi [1,3], Enrique Saez [1✉] & R. Luke Wiseman [1✉]

In obesity, signaling through the IRE1 arm of the unfolded protein response exerts both protective and harmful effects. Overexpression of the IRE1-regulated transcription factor XBP1s in liver or fat protects against obesity-linked metabolic deterioration. However, hyperactivation of IRE1 engages regulated IRE1-dependent decay (RIDD) and TRAF2/JNK pro-inflammatory signaling, which accelerate metabolic dysfunction. These pathologic IRE1-regulated processes have hindered efforts to pharmacologically harness the protective benefits of IRE1/XBP1s signaling in obesity-linked conditions. Here, we report the effects of a XBP1s-selective pharmacological IRE1 activator, IXA4, in diet-induced obese (DIO) mice. IXA4 transiently activates protective IRE1/XBP1s signaling in liver without inducing RIDD or TRAF2/JNK signaling. IXA4 treatment improves systemic glucose metabolism and liver insulin action through IRE1-dependent remodeling of the hepatic transcriptome that reduces glucose production and steatosis. IXA4-stimulated IRE1 activation also enhances pancreatic function. Our findings indicate that systemic, transient activation of IRE1/XBP1s signaling engenders multi-tissue benefits that integrate to mitigate obesity-driven metabolic dysfunction.

[1] Department of Molecular Medicine, The Scripps Research Institute, La Jolla, CA 92037, USA. [2] Department of Chemistry, The Scripps Research Institute, La Jolla, CA 92037, USA. [3] Department of Medicine, University of Wisconsin, Madison, WI 53705, USA. [4]These authors contributed equally: Aparajita Madhavan, Bernard P. Kok. ✉email: esaez@scripps.edu; wiseman@scripps.edu

The endoplasmic reticulum (ER) transmembrane protein IRE1 regulates the most evolutionarily conserved arm of the unfolded protein response (UPR)[1,2]. In response to ER stress, IRE1 is activated through a mechanism that involves autophosphorylation, oligomerization, and allosteric activation of its cytosolic RNase domain (Fig. 1a)[1,2]. Once activated, IRE1-signals via three mechanisms. IRE1 RNase activity promotes non-canonical splicing of *XBP1* mRNA that generates the stress-responsive transcription factor XBP1s[1,2]. Sustained IRE1 RNase activity also stimulates promiscuous degradation of mRNA and miRNA through a process called regulated IRE1-dependent decay (RIDD)[3,4]. Finally, when chronically activated, phosphorylated IRE1 recruits TRAF2 to initiate JNK-mediated pro-apoptotic signaling and NF-κB-regulated pro-inflammatory signaling[2,5–7].

The relative activity of these three IRE1-regulated signaling mechanisms is a critical determinant in dictating tissue-specific remodeling in the context of obesity and obesity-linked diseases such as type 2 diabetes[8–10]. IRE1-dependent XBP1s signaling functions primarily to protect tissues such as the liver and the pancreas from the stress brought about by chronic nutrient excess. Reduced translocation of XBP1s to the nucleus in the liver of obese mice, and the ensuing decrease in XBP1s transcriptional activity, contribute to impaired systemic glucose homeostasis[11]. Similarly, deletion of *Xbp1* in the liver or pancreas causes metabolic dysfunction, hepatic steatosis, insulin resistance, and impaired insulin secretion[12–16]. In contrast, genetic over-expression of the active XBP1s transcription factor to moderate, physiologically relevant levels in the liver of obese mice reduces hepatic gluconeogenesis and steatosis and improves systemic glucose metabolism[17–20]. Comparable improvements in systemic metabolism are observed when *Xbp1s* is overexpressed in other tissues such as adipose depots[21].

However, while XBP1s activity brought about by mild IRE1 activation is protective in obesity, prolonged IRE1 activation stimulates IRE1-dependent RIDD and JNK signaling, processes that hasten metabolic deterioration in obesity models[10]. For instance, IRE1-dependent JNK signaling promotes inhibitory phosphorylation of insulin receptor substrate 1 (IRS-1) and

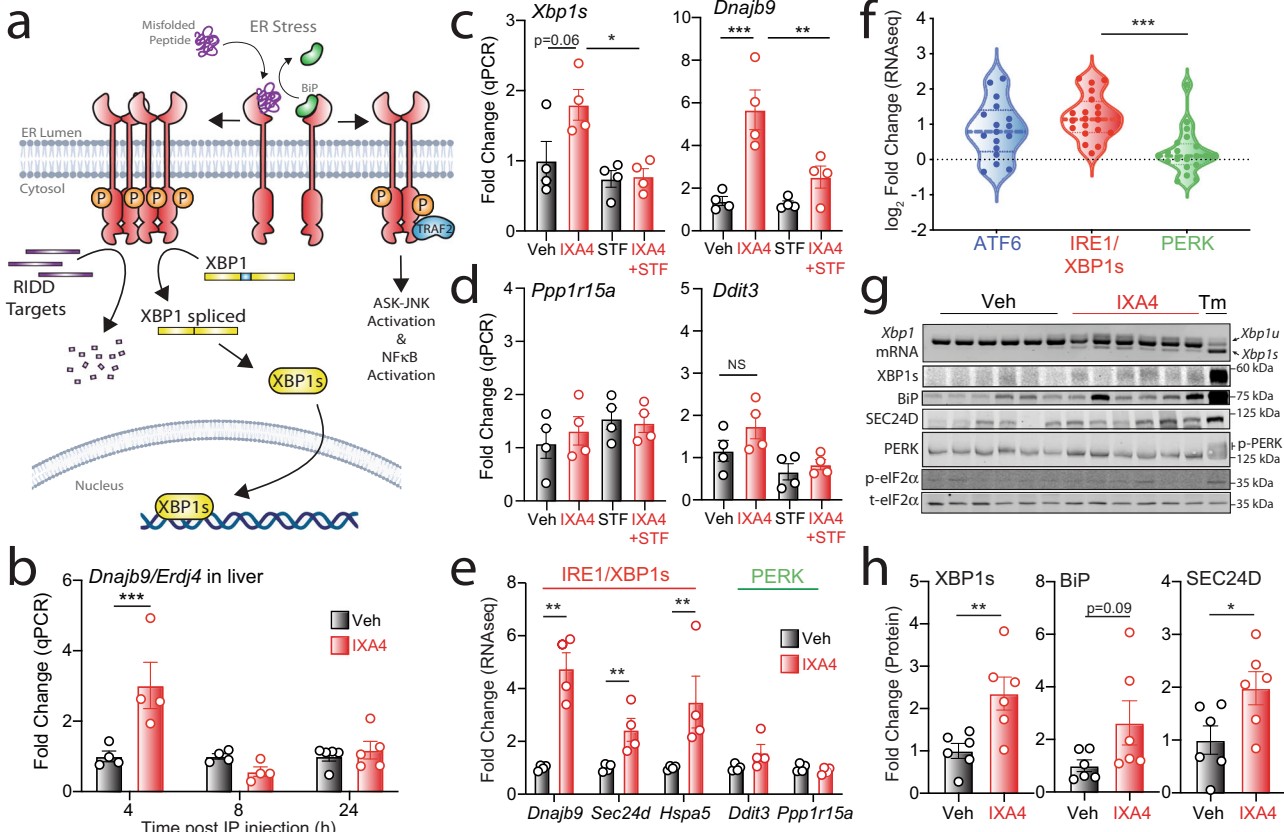

**Fig. 1 IXA4 selectively induces protective IRE1/XBP1s signaling in the liver. a** Schematic representation of the three downstream consequences of ER stress-induced IRE1 activation: XBP1s-dependent transcriptional signaling, RIDD-associated RNA degradation, and TRAF2-mediated ASK-JNK and NF-κB signaling. **b** Expression, measured by RT-qPCR, of the IRE1 target gene *Dnajb9* in the liver of chow-fed mice 4, 8, or 24 h after acute IXA4 intraperitoneal (IP) administration (50 mg/kg). Error bars show SEM for $n = 4$ or 5 mice. ***$P < 0.001$ for two-way ANOVA. **c, d** Expression, measured by RT-qPCR, of the IRE1-regulated genes *Xbp1s* and *Dnajb9* (**c**) or the PERK-regulated genes *Ppp1r15a* and *Ddit3* (**d**) in the liver of chow-fed mice 4 h after IP administration of IXA4 (50 mg/kg) and/or STF-083010 (STF; 10 mg/kg). Error bars show SEM for $n = 4$ mice. ***$P < 0.001$, **$P < 0.01$, *$P < 0.05$ for one-way ANOVA. **e** Fold change in IRE1/XBP1s and PERK target genes, measured by RNA-seq (Supplementary Data 1), in the liver of DIO mice after 8 weeks of IXA4 (50 mg/kg) treatment. Error bars show SEM for $n = 4$ mice. **$P < 0.01$ for two-way ANOVA. **f** Fold change of ATF6, IRE1, and PERK target genesets[31] in the liver of DIO mice after 8 weeks of IXA4 treatment assessed by RNA-seq. Data are shown normalized to vehicle. Specific genes used in this analysis are identified in Supplementary Data 2. ***$P < 0.001$ for one-way ANOVA. **g, h** *Xbp1* mRNA gel (top) and immunoblots of UPR-regulated proteins in the liver of DIO mice after 8 weeks of IXA4 (50 mg/kg) treatment (**g**). Liver from mice treated for 16 h with tunicamycin (Tm; 1 mg/kg IP administered) is shown as a control. Quantification of XBP1s, BiP, and SEC24D in these immunoblots is shown in (**h**). Error bars show SEM for $n = 6$ mice. **$P < 0.01$, *$P < 0.05$ for a two-tailed unpaired *t* test. Source data for all panels in this figure are provided as Source Data File 1. Uncropped images of the immunoblots in panel (**g**) are included in Source Data.

suppresses insulin signaling in hepatocytes[8]. Similarly, hyper-activation of IRE1 in the liver of obese mice is associated with pro-inflammatory signaling and has been implicated in the pathogenesis of hepatocellular carcinoma[22,23]. Chronic IRE1 activity also promotes metabolic dysfunction in other tissues such as the pancreas, where sustained IRE1 signaling is associated with both, β-cell apoptosis and RIDD-dependent degradation of insulin mRNA[10,24].

The detrimental role of IRE1 hyperactivity in metabolic disease has led to the development of pharmacologic strategies to inhibit IRE1 signaling in obesity and related conditions[25,26]. However, because most IRE1 inhibitors also block the increase in XBP1s signaling that protects tissues from the effects of nutrient over-load, the extent to which IRE1 inhibitors may be useful in obesity remains to be established. An alternative pharmacological approach to mitigate metabolic dysfunction in obesity might be to selectively enhance protective IRE1/XBP1s signaling without inducing the pathologic aspects associated with IRE1 hyper-activity (e.g., TRAF2-mediated JNK activation)[10,25,26]. We recently identified a compound, IXA4, that activates protective IRE1/XBP1s signaling in mammalian cells through a mechanism requiring IRE1 autophosphorylation[27]. Interestingly, this com-pound does not stimulate RIDD or TRAF2-mediated JNK or NF-κB signaling, likely reflecting its ability to induce only a transient, modest activation of IRE1[27]. As such, IXA4 offers a heretofore unprecedented opportunity to evaluate the extent to which pharmacological activation of XBP1s signaling might correct metabolic dysfunction in the context of obesity and its associated conditions.

In this work, we show that IXA4 activates protective IRE1/XBP1s signaling in the liver of obese insulin-resistant DIO mice without inducing RIDD, TRAF2/JNK signaling, or associated pathologies, such as liver inflammation and fibrosis. Chronic pharmacological activation of protective IRE1/XBP1s signaling reduces hepatic gluconeogenesis and liver steatosis in an IRE1-dependent manner, thus enhancing systemic metabolic function. IXA4 treatment also enhances pancreatic β-cell function and nutrient-stimulated insulin secretion. Our findings indicate that simultaneous activation of IRE1/XBP1s signaling in key metabolic tissues (e.g., liver, pancreas) brought about by treatment with a selective pharmacological activator such as IXA4, stimulates a beneficial remodeling in multiple tissues that, in aggregate, mitigates systemic metabolic dysfunction.

## Results

**IXA4 stimulates IRE1/XBP1s signaling in the liver of chow-fed mice.** As a first step to define the potential of IXA4 to improve metabolic parameters in obesity, we treated primary mouse hepa-tocytes with IXA4 and measured expression of *Xbp1s* and the XBP1s target gene *Dnajb9/Erdj4*. IXA4 treatment induced expression of both of these transcripts (Supplementary Fig. S1a–c). Co-treatment with two structurally distinct IRE1 RNAse inhibitors, 4μ8c[28] or STF-083010[29], blocked compound-stimulated *Xbp1* splicing and induction of XBP1s targets such as *Dnajb9*, confirming that IXA4 acted in an IRE1-dependent manner. Next, to test if similar effects would be observed in mice, we dosed chow-fed mice with IXA4 (50 mg/kg; intraperitoneal injection) and monitored the expression of the XBP1s target gene *Dnajb9* at multiple time points. IXA4 treatment increased hepatic expression of *Dnajb9* at 4 h, with levels returning to baseline within 8 h of dosing (Fig. 1b). Expression of *Xbp1s* was similarly increased in liver 4 h after administration of IXA4 (Fig. 1c). Co-administration of STF-083010 (10 mg/kg; intraperitoneal injection) inhibited IXA4-induced increases in *Dnajb9* and *Xbp1s*, demonstrating that IXA4 stimulates their expression through an IRE1-dependent mechanism (Fig. 1c). This

finding is consistent with the observation that IXA4 does not acti-vate IRE1/XBP1s signaling in *Ire1*-deficient cells[27]. We did not detect increased liver expression of genes regulated by the PERK UPR signaling pathway (e.g., *Ddit3/Chop*, *Ppp1r15a/Gadd34*) 4 h after IXA4 treatment, indicating that this UPR pathway was not activated (Fig. 1d). We did note increased expression of *Hspa5/BiP*, a gene primarily regulated by the ATF6 arm of the UPR[30,31], 4 h after administration of IXA4 (Supplementary Fig. S1d). *Hspa5* is also induced by XBP1s, albeit to a lower extent to that observed with ATF6[30,31], suggesting that increased expression of this gene in IXA4-treated mice could be attributed to IRE1/XBP1s activation. Consistent with this notion, expression of *Hspa5* in IXA4-treated *Atf6*[+/+] and *Atf6*[−/−] MEFs[32] was similar (Supplementary Fig. S1e). Moreover, co-treatment of primary hepatocytes with the IRE1 inhibitor 4μ8c blunted the IXA4-induced increase in *Hspa5* expression (Supplementary Fig. S1f). Similarly, the increase in *Hspa5* expression seen in the liver of chow-fed IXA4-treated mice 4 h post-dosing was reduced by co-administration of the IRE1 inhibitor STF-083010 (Supplementary Fig. S1g). Collectively, these observations show that IXA4 treatment induced transient and preferential acti-vation of IRE1/XBP1s signaling in the liver, thus offering an opportunity to test the impact of pharmacologic IRE1/XBP1s acti-vation in the context of obesity.

**IXA4 selectively activates IRE1/XBP1s signaling in the liver of DIO mice.** To evaluate the therapeutic promise of increased IRE1/XBP1s activity, we treated diet-induced obese (DIO) mice with IXA4 (50 mg/kg daily) for 8 weeks. We then isolated their livers and assessed IRE1/XBP1s activation by RNA-seq, real-time (RT)-qPCR, and immunoblotting. Akin to what was observed in chow-fed mice, IXA4 treatment induced expression of IRE1/XBP1s target genes (e.g., *Dnajb9*, *Sec24d*) in DIO mice (Fig. 1e, f, Supplementary Data 1, and Supplementary Data 2). Gene set enrichment analysis (GSEA) con-firmed activation of the IRE1/XBP1s transcriptional program in IXA4-treated DIO mice (Supplementary Fig. S1h). In contrast, IXA4 treatment did not induce genes regulated by the PERK arm of the UPR, such as *Chop/Ddit3* or *Gadd34/Ppp1r15a* (Fig. 1e, f). Genes primarily regulated by ATF6 signaling (e.g., *Hspa5*) were modestly induced by IXA4 treatment, a reflection of the intrinsic, natural overlap between the XBP1s and ATF6 genesets[30,31] noted above. Importantly, co-administration of IXA4 and STF-083010 (5 mg/kg) daily for 3 weeks attenuated IXA4-stimulated expression of *Dnajb9* and *Hspa5* in the liver of DIO mice, confirming that IXA4 treatment primarily induced *Hspa5* expression by increasing IRE1/XBP1s activity and not ATF6 signaling (Supplementary Fig. S1i). GSEA analysis revealed that IXA4 treatment did not activate any additional stress-responsive transcriptional pathways, such as the heat shock response (HSR) or the oxidative stress response (OSR) (Supple-mentary Fig. S1j, k). Gene ontology (GO) also indicated that path-ways induced by IXA4 treatment were largely associated with UPR-linked processes, including ER stress and secretory proteostasis (Supplementary Data 3), offering further evidence of the selectivity of this molecule in the liver of DIO mice. As expected, IXA4 treatment also increased the levels of nuclear XBP1s protein and of XBP1s-regulated proteins (e.g., BiP, SEC24D) (Fig. 1g, h). In contrast, IXA4 treatment did not induce PERK phosphorylation or phosphorylation of the PERK kinase substrate eIF2α, corroborating that IXA4 did not activate this pathway in the liver of DIO mice. These transcriptional and immunoblotting results indicate that, in the liver of treated DIO mice, IXA4 preferentially activated the IRE1/XBP1s signaling arm of the UPR.

Despite the noted increase in IRE1/XBP1s signaling, chronic treatment with IXA4 did not alter the expression of hepatic RIDD targets (Supplementary Fig. S1)[33]. Similarly, IXA4 treatment did not increase JNK phosphorylation or downstream induction of

pro-apoptotic genes in the liver of DIO mice (Supplementary Fig. S1m, n). We also did not observe increases in NF-κB transcriptional activity (Supplementary Fig. S1o). These results indicate that IXA4 did not stimulate RIDD or TRAF2-mediated IRE1 signaling in the liver of treated DIO mice (Fig. 1a). This finding likely reflects the transient nature of IXA4-induced IRE1/ XBP1s activation in the liver (Fig. 1b), which is expected to limit the deleterious effects linked to sustained IRE1 signaling. Although we noticed an upregulation of select acute phase response (e.g., *Saa1*, *Saa2*, and *Orm2*) and fibrosis genes (e.g., *Col1a1* and *Timp1*) in livers of IXA4-treated DIO mice (Supplementary Fig. S1p, q), IXA4 treatment did not increase liver fibrosis (Supplementary Fig. S1r) or the levels of multiple plasma cytokines (Supplementary Fig. S1s). In addition, no increase in mRNA levels of the macrophage marker *Cd68* or of ceramide synthesis genes associated with IRE1-stimulated hepatocyte inflammation[34] was seen (Supplementary Fig. S1t, u). These results indicate that IXA4 treatment did not exacerbate hepatic inflammation or fibrosis in DIO mice. Consistent with this notion, IXA4 treatment did not alter ALT or AST plasma levels following 3 weeks of treatment (Supplementary Fig. S1v). After 8 weeks of IXA4 treatment, plasma ALT levels were reduced, an indication of improved liver function (Supplementary Fig. S1w). Together, these findings show that IXA4 treatment stimulated protective IRE1/XBP1s signaling in the liver of DIO mice in the absence of pathologic IRE1 activities.

**Treatment with IXA4 improves glucose homeostasis in DIO mice.** Liver overexpression of the active XBP1s transcription factor to physiologically relevant levels improves systemic glucose metabolism in obese-diabetic mice[17,18,20]. Because IXA4 treatment preferentially activated protective IRE1/XBP1s signaling in the liver, we anticipated that this compound would similarly improve systemic glucose metabolism in DIO mice. Indeed, IXA4 treatment reduced fasting blood glucose, plasma insulin, and the HOMA-IR in DIO mice, reflecting improvements in insulin sensitivity (Fig. 2a–c). IXA4 treatment also increased systemic glucose clearance measured using a glucose tolerance test (GTT) when glucose was administered either orally (OGTT; Fig. 2d, e) or intraperitoneally (IPGTT; Fig. 2f, g). IXA4-induced enhancement of glucose clearance was independent of changes in body weight or food intake, which were not different between treatment groups (Supplementary Fig. S2a, b). Levels of plasma triglycerides and free fatty acids were also not altered by IXA4 treatment, though there was a modest decrease in plasma cholesterol in treated mice (Supplementary Fig. S2c–e). Importantly, the improvement in glucose tolerance seen in IXA4-treated DIO mice was not observed in mice co-treated with IXA4 and STF-083010, indicating that this benefit of IXA4 treatment can be ascribed to IRE1 activity (Fig. 2h). As was reported previously in the case of hepatic XBP1s overexpression[18], we noted no increase in insulin-stimulated glucose clearance, as assessed in an insulin tolerance test (ITT), in DIO mice treated with IXA4 for 2 weeks (Fig. 2i). AKT phosphorylation was also not affected in gastrocnemius muscle isolated from IXA4-treated DIO mice and acutely challenged ex vivo with insulin (Supplementary Fig. S2f). Similarly, AKT phosphorylation was not altered in quadriceps muscle of IXA4-treated DIO mice isolated 15 min after an in vivo oral glucose bolus (Supplementary Fig. S2g). However, AKT phosphorylation was increased in livers isolated from IXA4-treated DIO mice after oral administration of a glucose bolus or a meal challenge (Fig. 2j, k and Supplementary Fig. S2h), reflecting improved insulin action in this tissue. These observations indicate that IXA4 treatment improved glucose homeostasis in the liver of treated DIO mice in physiologically relevant contexts such as feeding.

**IXA4 treatment suppresses hepatic gluconeogenesis.** A primary mechanism whereby overexpression of XBP1s in the liver enhances hepatic glucose control is the induction of post-translational degradation of FOXO1—a key transcriptional regulator of hepatic gluconeogenic gene expression[18]. We found that treatment of primary mouse hepatocytes with IXA4 reduced FOXO1 protein levels independent of changes in *Foxo1* mRNA (Supplementary Fig. S3a, b). The IXA4-induced decrease in FOXO1 levels was attenuated by co-treatment with the IRE1 inhibitor 4μ8c, indicating that it requires IRE1 signaling. Consistent with decreased FOXO1 activity, IXA4 treatment reduced protein levels of the FOXO1-regulated gluco-neogenic enzyme PCK1 in primary hepatocytes (Supplementary Fig. S3c). Co-treatment with 4μ8c restored PCK1 levels in these cells, showing that this reduction could be attributed to increased IRE1 activity. IXA4 treatment also tended to decrease FOXO1 protein levels in the liver of chronically treated DIO mice (Supplementary Fig. S3d), though in this setting the reduction in FOXO1 levels was accompanied by a decrease in *Foxo1* mRNA, probably reflecting the consequences of chronic, long-term treatment with an IRE1/XBP1s activator (Supplementary Fig. S3e). Likely as a result of reduced FOXO1 activity, chronic IXA4 treatment broadly decreased expression of gluconeogenic genes in the liver of DIO mice (Fig. 3a, Supplementary Fig. S3f, and Supplementary Data 4). Specifically, expression of the key gluconeogenic genes *Pck1* and *G6pc* was reduced in the liver of DIO mice treated with IXA4 for 8 weeks (Fig. 3b). Similar results were observed in livers of DIO mice treated with IXA4 for 3 weeks (Fig. 3c). Co-administration of STF-083010 inhibited IXA4-induced suppression of hepatic gluconeogenic gene expression (Fig. 3c). As might be expected given these observations, hepatic glucose production, assessed in a pyruvate-tolerance test (PTT), was reduced in IXA4-treated DIO mice (Fig. 3d). These findings indicate that IXA4 treatment stimulated adaptive remodeling of the liver to limit hepatic gluconeogenesis.

**Treatment with IXA4 reduces hepatic steatosis.** XBP1s also plays an anti-lipogenic role in the liver, suppressing the expression of lipid metabolism genes and reducing liver steatosis[17]. Accordingly, IXA4 treatment decreased liver expression of key lipogenic genes, including *Dgat2*, *Scd1*, and *Srebf1c*, in DIO mice treated chronically (Fig. 3e and Supplementary Fig. S3g, h). Co-treatment with STF-083010 inhibited IXA4-induced suppression of these genes, indicating that IXA4 reduces their levels in an IRE1-dependent manner (Fig. 3e). Similar results were observed in primary hepatocytes treated with IXA4 and either 4μ8c or STF-083010 (Supplementary Fig. S3i, j). This suppression of lipogenic gene expression was reflected in a reduction in liver triglyceride content and decreased steatosis in DIO mice treated chronically with IXA4 (Fig. 3f, g). No changes were seen in liver cholesterol content (Supplementary Fig. S3k). While these effects were more robust following 8 weeks of treatment, they were also evident in DIO mice treated with IXA4 for only 3 weeks: IXA4 treatment decreased hepatic triglyceride content and steatosis, and these benefits disappeared in mice dosed with both IXA4 and STF-083010 (Supplementary Fig. S3l, m). These findings suggest that, in addition to decreasing hepatic glucose production, IXA4 treatment also caused adaptive IRE1-dependent remodeling of liver lipid metabolism to reduce steatosis in DIO mice.

**IXA4 improves glucose-stimulated insulin secretion.** An advantage of systemic administration of pharmacologic IRE1/ XBP1s activators such as IXA4 is the potential to stimulate

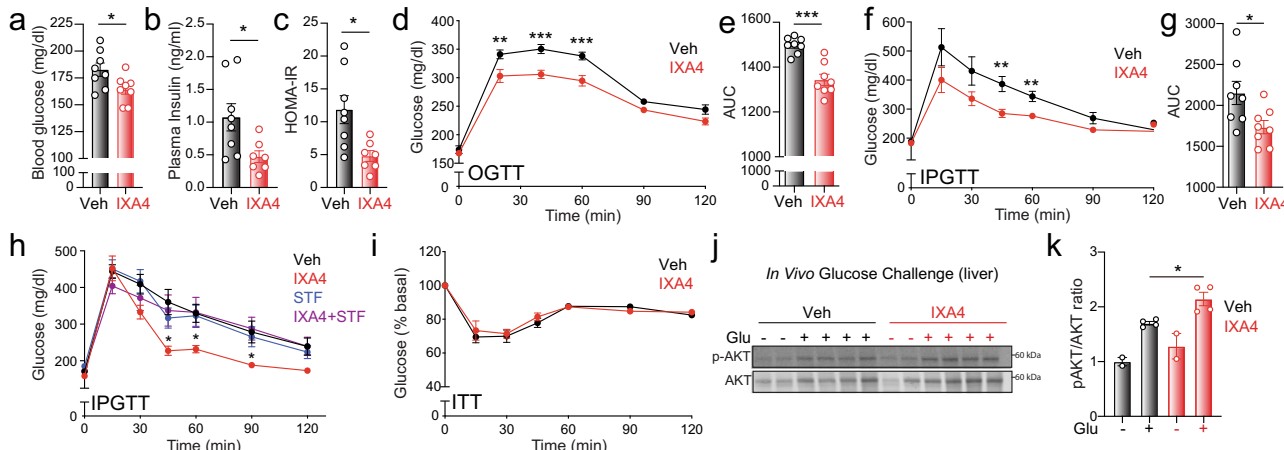

**Fig. 2 Chronic IXA4 treatment enhances systemic glucose homeostasis. a–c** Fasted plasma glucose (**a**), insulin (**b**), and HOMA-IR (**c**) in DIO mice after 46 days of IXA4 treatment. Error bars show SEM for $n = 7$ or 8 mice. One IXA4-treated mouse was excluded from panels **b**, **c** using a ROUT outlier test. *$P < 0.05$ for a two-tailed Welch's $t$ test. **d** Oral GTT in DIO mice 38 days after start of IXA4 treatment. Error bars show SEM for $n = 8$ mice/condition. **$P < 0.01$, ***$P < 0.001$ for two-way ANOVA. **e** Area under the curve (AUC) quantification of the OGTT in (**d**). ***$P < 0.001$ for a two-tailed Welch's $t$ test. **f** Intraperitoneal GTT (IPGTT) in DIO mice 22 days after start of IXA4 treatment. Error bars show SEM for $n = 8$ mice/condition. **$P < 0.01$ for a two-tailed Welch's $t$ test. **g** AUC quantification of the IPGTT in (**f**). *$P < 0.05$ for a two-tailed Welch's $t$ test. **h** IPGTT in DIO mice 10 days after start of treatment with IXA4 (50 mg/kg) and/or STF-083010 (STF; 5 mg/kg). Error bars show SEM for $n = 8$ mice/condition. *$P < 0.05$ for two-way ANOVA comparing mice treated with IXA4 in the presence or absence of STF. **i** ITT in DIO mice after 17 days of IXA4 treatment. Error bars show SEM for $n = 8$ mice/condition. **j** Immunoblot of AKT phosphorylation in liver 15 min after an oral bolus of glucose was administered to DIO mice treated for 60 days with IXA4. **k** pAKT/AKT ratio of immunoblots shown in (**j**). Error bars show SEM for $n = 2$ or 4 replicates, *$P < 0.05$ for a two-tailed Welch's $t$ test. Source data for all panels in this figure are provided as Source Data File 2. Uncropped images of the immunoblots in panel (**j**) are included in Source Data.

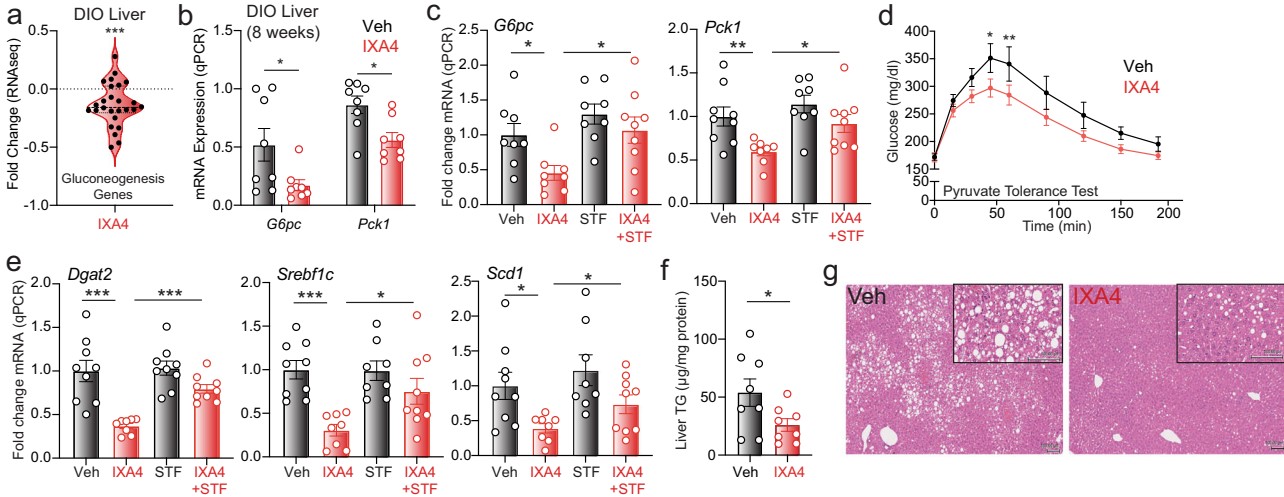

**Fig. 3 IXA4 improves liver function in DIO mice. a** Fold change, assessed with RNA-seq, of gluconeogenic genes in the liver of DIO mice treated with IXA4 for 8 weeks. Genes used in this analysis are shown in Supplementary Data 3. ***$P < 0.001$ for a two-tailed unpaired $t$ test comparing expression of genes to an average of livers from vehicle-treated DIO mice. **b** Expression, measured by RT-qPCR, of gluconeogenic genes $G6pc$ and $Pck1$ in the liver of DIO mice treated with IXA4 for 8 weeks. Error bars show SEM for $n = 8$ mice/condition. *$P < 0.05$ for a two-tailed Welch's $t$ test. **c** Expression, measured by RT-qPCR, of $G6pc$ and $Pck1$ in the liver of DIO mice treated for 21 days with IXA4 and/or STF-083010 (STF; 5 mg/kg). Error bars show SEM for $n = 8$ or 9 mice/condition. *$P < 0.05$, **$P < 0.01$ for a two-tailed Welch's $t$ test. **d** Pyruvate-tolerance test (PTT) in DIO mice treated with IXA4 for 22 days. Error bars show SEM for $n = 9$ mice/condition. *$P < 0.05$, **$P < 0.01$ for two-way ANOVA. **e** Expression, measured by qPCR, of the lipogenic genes $Dgat2$, $Srebf1c$, and $Scd1$ in the liver of DIO mice treated for 21 days with IXA4 and/or STF-083010 (STF; 5 mg/kg). Error bars show SEM for $n = 8$ or 9 mice/condition. *$P < 0.05$, ***$P < 0.001$ for a two-tailed Welch's $t$ test. **f** Triglyceride content in liver of DIO mice treated with IXA4 for 8 weeks. Error bars show SEM for $n = 8$ mice/condition. *$P < 0.05$ for a two-tailed unpaired $t$ test. **g** Representative liver images of DIO mice (from $n = 8$ images/condition) treated with IXA4 for 8 weeks, stained with H&E. Source data for all panels in this figure are provided as Source Data File 3.

protective IRE1 signaling in multiple tissues. The reductions in fasting plasma glucose and insulin observed in IXA4-treated DIO mice (Fig. 2a, b), together with the improved ability of treated mice to clear glucose during a GTT (Fig. 2d, h), suggested that

IXA4 treatment might have also enhanced pancreatic β-cell function. Hence, we assessed the effect of IXA4 treatment on the pancreas of IXA4-treated DIO mice. Chronic IRE1 hyperactivation in the pancreas alters insulin secretion by promoting islet

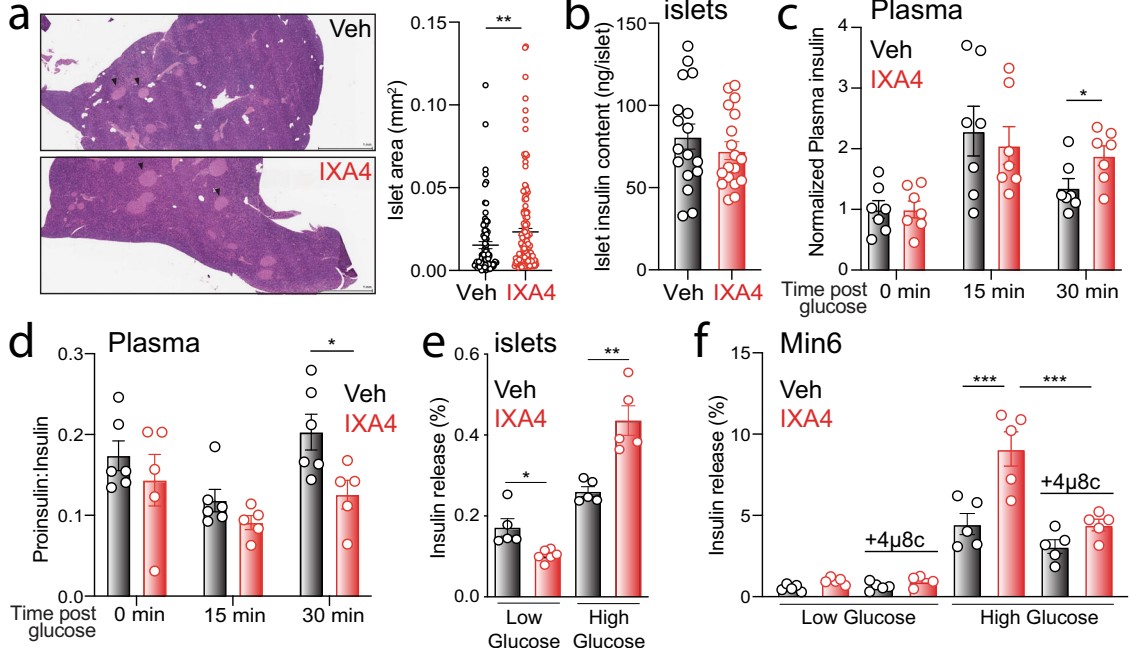

**Fig. 4 IXA4 treatment enhances pancreatic function. a** Representative images and quantification of islet area in the pancreas of DIO mice treated with IXA4 for 8 weeks and stained with H&E. Arrowheads indicate individual islets. The mean islet area is shown and error bars represent SEM for $n = 84$ or 143 replicates. **$P < 0.01$ for a two-tailed Welch's $t$ test. **b** Insulin content of primary islets isolated from DIO mice after 8 weeks of IXA4 treatment. Error bars show SEM for $n = 18$ replicates. **c** Time course of plasma insulin levels after oral administration of a glucose bolus normalized to basal insulin at time 0 h. Experiment conducted after 38 days of IXA4 treatment. Error bars show SEM for $n = 7$ mice/condition. *$P < 0.05$ for a two-tailed Welch's $t$ test. **d** Plasma proinsulin to insulin ratio after oral administration of a glucose bolus to DIO mice treated for 38 days with IXA4. Error bars show SEM for $n = 5$ or 6 mice. *$P < 0.05$ for two-way ANOVA. **e** Percent insulin release after 60 min of low and high glucose stimulation from primary islets isolated from DIO mice treated for 8 weeks with IXA4. Error bars show SEM for $n = 5$ or 6 mice/condition. Outliers identified by the Grubbs' test were removed from this analysis. *$P < 0.05$, **$P < 0.01$ for a two-tailed Welch's $t$ test. **f** Insulin release from Min6 cells pretreated for 36 h with vehicle, IXA4 (10 μM), and/or 4μ8c (32 μM) and then stimulated with media containing low (2.8 mM) or high (16.8 mM) glucose for 60 min. Error bars show SEM for $n = 5$ replicates. ***$P < 0.001$ for one-way ANOVA. Source data for all panels in this figure are provided as Source Data File 4.

apoptosis and RIDD-dependent insulin mRNA degradation[10,24]. However, we observed no change in pancreas morphology or islet number in IXA4-treated DIO mice, though we did notice a modest increase in islet size (Fig. 4a). IXA4 treatment did not alter insulin content in primary pancreatic islets isolated from treated mice (Fig. 4b). Together, these findings indicate that IXA4 treatment had no major impact on β-cell proliferation or apoptosis, or on islet insulin content, confirming that, as was the case in the liver, IXA4 treatment did not increase pathologic IRE1 signaling in the pancreas.

Interestingly, DIO mice chronically treated with IXA4 displayed a more sustained increase in plasma insulin following an oral glucose bolus (Fig. 4c). Further, the proinsulin:insulin ratio in plasma was reduced in IXA4-treated mice, indicating greater processing of proinsulin to mature, active insulin following glucose challenge (Fig. 4d). These findings pointed to enhanced glucose-stimulated insulin secretion (GSIS) in IXA4-treated DIO mice. Consistent with this notion, basal insulin secretion was reduced and GSIS was increased in primary pancreatic islets isolated from IXA4-treated DIO mice (Fig. 4e). IXA4-induced increases in GSIS were also observed in Min6 cells (Fig. 4f). Co-treatment with the IRE1 RNAse inhibitor 4μ8c, which blocks IRE1-dependent *Xbp1* splicing in Min6 cells (Supplementary Fig. S4a), reversed IXA4-induced enhancement of GSIS (Fig. 4f). IXA4-induced enhancement of GSIS was also abolished in Min6 cells treated with STF-083010 (Supplementary Fig. S4b) or bearing genetic depletion of *Ire1* (Supplementary

Fig. S4c, d), confirming that this effect requires IRE1 activity. Importantly, we observed no reduction in *Ins1* or *Ins2* mRNA levels in IXA4-treated Min6 cells, indicating that this compound does not induce RIDD-dependent degradation of insulin mRNA in these cells (Supplementary Fig. S4e). These results show that IXA4 treatment did not spur pancreatic dysfunction associated with chronic IRE1 hyperactivity, but rather, it induced an adaptive IRE1-dependent reprogramming of pancreatic β-cell function that improved insulin regulation and secretion in the pancreas of treated DIO mice.

**Concluding remarks**
Our findings reveal the potential of pharmacologically and selectively increasing IRE1/XBP1s signaling using compounds like IXA4 to stimulate a beneficial reprogramming in several tissues to mitigate obesity-driven metabolic dysfunction. By selectively activating protective IRE1-dependent XBP1s signaling in a transient fashion, IXA4 induced adaptive remodeling of multiple tissues in obese insulin-resistant mice without eliciting the deleterious effects associated with chronic IRE1 hyperactivation. Mirroring what was observed upon genetic overexpression of XBP1s in the liver[17,18], IXA4-stimulated IRE1 activation remodeled liver gene expression to reduce hepatic gluconeogenesis and ectopic lipid deposition. IXA4 treatment also improved insulin regulation and secretion in the pancreas, uncovering a new mechanism through which pharmacologic IRE1 activation

aids to restore homeostasis in metabolic disease. These findings provide a compelling rationale for the continued development and application of IRE1/XBP1s-activating compounds to mitigate pathologies associated with obesity and other complex diseases[25,26].

## Methods

**Chemicals and antibodies**. The IRE1/XBP1s activator IXA4 was custom synthesized by Otava chemicals. The IRE1 RNase inhibitor 4µ8c was purchased from EMD Millipore (catalog no. 412512) and STF-083010 was purchased from Selleck Chemicals (catalog no. S7771). Antibodies used in this study include: phospho-c-jun (Cell Signaling, catalog no. 3270S), phospho-JNK (Cell Signaling, catalog no. 4668S), JNK (Cell Signaling, catalog no. 9252S), phospho-AKT(Ser473) (Cell Signaling, catalog no. 4060S), AKT (Cell Signaling, catalog no. 2920S), FOXO1 (Cell Signaling, catalog no. 2880S), LaminB1 (Cell Signaling, catalog no. 13435S), tubulin (Sigma, catalog no. T6074-200UL), XBP1s (E9V3E) (Cell Signaling, catalog no. 40435S), BiP (Cell Signaling, catalog no. 3177S), SEC24D (gift from William Balch's lab at Scripps), PERK (Cell Signaling, catalog no 3192S), eIF2α (Cell Signaling, catalog no 9722), phospho-eIF2α (Cell Signaling, catalog no 9721), and PCK1 (Abcam, catalog no. ab70358).

**Cell culture**. Min6 cells (a kind gift from Maike Sander at UCSD) were cultured in DMEM supplemented with 10% FBS, 2 mM glutamine, 0.05 mM β-mercaptoethanol, 100 U ml$^{-1}$ penicillin, and 100 µg ml$^{-1}$ streptomycin in standard culture conditions (37 °C, 5% $CO_2$). Primary hepatocytes were isolated from 2-month-old male C57BL/6J mice, as previously described[35]. Briefly, we cannulated the hepatic portal vein of anesthetized mice and perfused the liver with warm (37 °C) Hank's balanced salt solution containing 25 mM HEPES and 0.5 mM EGTA for 5 min, followed by 0.1% collagenase type IV in Hank's balanced salt solution containing 25 mM HEPES and 5 mM $CaCl_2$ for another 5 min. We then removed the perfused liver, agitated it, and dispersed the cells into culture medium. We filtered the resulting cell suspension through a fine-mesh strainer (100 µm), and washed cells twice with culture medium. We selected viable cells using Percoll centrifugation and resuspended them in culture medium before plating them in collagen I-coated plates. Primary hepatocytes were cultured in DMEM supplemented with 10% FBS, 2 mM glutamine, 100 U ml$^{-1}$ penicillin, and 100 µg ml$^{-1}$ streptomycin in standard culture conditions (37 °C, 5% $CO_2$). Lentivirus for *Ire1* knockdown in Min6 cells was prepared in HEK293T cells (ATCC CRL-3216) cultured in DMEM supplemented with 10% FBS, 2 mM glutamine, 100 U ml$^{-1}$ penicillin, and 100 µg ml$^{-1}$ streptomycin in standard culture conditions (37 °C, 5% $CO_2$). These cells were transfected using calcium phosphate with a MISSION *Ire1* lentiviral shRNA construct (Sigma Aldrich) and the viral packaging plasmids VSV-G, RRE, and REV. The supernatant containing viral particles was collected at 24 and 48 h post transfection, concentrated threefold, and applied onto Min6 cells to induce *Ire1* knockdown. Min6 cells were allowed to grow for 48 h in virus-containing medium and a GSIS assay was subsequently performed.

**Mouse studies**. Chow-fed male C57BL/6J mice (10-week-old) were purchased from the Scripps Research breeding colony. Male C57BL/6N diet-induced obese (DIO) mice (14-week-old) were purchased from Taconic Biosciences. DIO mice were maintained on a 60% kcal high-fat diet (Research Diets D12492) for 3 weeks prior to IXA4 treatment. IXA4 was formulated in 10% DMSO, 30% Kolliphor EL:ethanol (2:1 ratio), 60% saline and kept warm until use. STF-083010 was formulated in 10% DMSO, 16% Kolliphor EL:ethanol (2:1 ratio), and saline and kept warm until use. Chow-fed or DIO mice were administered vehicle, IXA4 (50 mg/kg), and/or STF-083010 (5 or 10 mg/kg, as indicated) once daily via intraperitoneal injection for up to 8 weeks, as indicated in the text. Body weight and food intake were measured weekly throughout the studies. At sacrifice, tissues were harvested and flash-frozen for processing, or collected in formalin for histology. Mouse experiments were approved by and conducted in accordance with the guidelines of The Scripps Research Institute IACUC.

**Glucose, insulin, and pyruvate-tolerance tests**. For glucose-tolerance tests, mice treated with vehicle, IXA4, and/or STF-083010 were fasted overnight for 12 h (21:00 to 09:00) prior to administration of a 2 g/kg glucose bolus via oral gavage or intraperitoneal injection, as noted in the text. For insulin-tolerance tests, mice were fasted for 4 h (09:00 to 13:00) prior to intraperitoneal injection of Novolin insulin (0.75 U/kg). For pyruvate-tolerance tests, mice were fasted overnight for 12 h (21:00 to 09:00) prior to intraperitoneal injection of sodium pyruvate (1 g/kg). In all cases, plasma glucose levels were determined at the indicated time points using the Clarity BG1000 blood glucose monitoring system (Clarity Diagnostics).

**In vivo insulin signaling upon glucose and meal challenge**. To measure insulin signaling upon administration of a glucose bolus or a complex meal, mice were administered 2 g/kg glucose or 200 mL Ensure Plus (catalog no. RS58303), respectively, via oral gavage following a 12 h (21:00 to 09:00) overnight fast. Mice were euthanized 15 min post glucose bolus or meal administration and tissues rapidly excised and frozen for processing and immunoblotting.

**Ex vivo insulin signaling**. For measuring insulin signaling upon insulin stimulation ex vivo, mice fasted overnight were euthanized and the soleus and gastrocnemius muscles rapidly harvested. Paired tissues were incubated in a modified Krebs-Henseleit buffer (138 mM NaCl, 4.7 mM KCl, 1.2 mM $MgSO_4$, 1.25 mM $CaCl_2$, 1.2 mM $KH_2PO_4$, 25 mM HEPES, and 11 mM glucose) supplemented with 0.1% BSA, 2 mM sodium pyruvate, and 6 mM mannitol and containing either vehicle or 10 nM insulin and incubated for 10 min with shaking. Tissues were then washed with PBS and flash-frozen in liquid nitrogen until processing and immunoblotting.

**Plasma measurements**. Blood glucose levels were determined using the Clarity BG1000 blood glucose monitoring system (Clarity Diagnostics). Blood was obtained by tail vein collection, retro-orbital collection from isoflurane-anesthetized mice, or by cardiac puncture post euthanasia. Plasma insulin was quantified using the Ultra Sensitive Mouse Insulin Elisa (Crystal chem; catalog no. 90080). Plasma proinsulin was measured using the Mercodia Rat/Mouse Proinsulin ELISA (catalog no. 10-1232-01). Plasma cytokines and chemokines were measured using the Bio-Rad Bio-Plex Pro Mouse Cytokine 23-plex Assay (catalog no. M60009RDPD). Serum triglycerides were quantified using the EnzyChrom Triglyceride Assay (BioAssay systems; catalog no. ETGA-200). Serum-free fatty acids were measured using the Free Fatty Acid Quantitation Kit (Sigma Aldrich, catalog no MAK044). Serum cholesterol was quantified using the Cholesterol E kit (Wako Diagnostics, catalog no 999-02601). Plasma ALT levels were measured using the Amplite Colorimetric Alanine Aminotransferase Assay Kit (AAT Bioquest; catalog no. 13803). Plasma AST levels were quantified using a colorimetric activity assay kit (Cayman Chemical; catalog no. 701640). HOMA-IR was calculated using the formula: HOMA-IR = glucose (mg/dl) × insulin (mU/L)/405.

**Primary pancreatic islet isolation and GSIS**. Primary mouse islets were isolated from DIO mice treated with IXA4 or vehicle for 8 weeks, as described previously[36]. Briefly, collagenase P (Roche) was perfused into the pancreas at a concentration of 0.8 mg/ml through the common bile duct. The pancreas was then removed and dissociated at 37 °C for a maximum of 15 min. Islets were separated using a gradient composed of HBSS and Histopaque (Sigma) layers. Purified islets were hand-picked using a dissection microscope. Islets from vehicle- or IXA4-treated mice were incubated overnight in RPMI 1640 supplemented with 8 mM glucose, 10% FBS, 2 mM L-glutamine, 100 U/mL Pen/Strep, 1 mM sodium pyruvate, 10 mM HEPES. Next day, islets were washed and pre-incubated for 1 h in Krebs-Ringers-Bicarbonate-HEPES (KRBH) buffer (130 mM NaCl, 5 mM KCl, 1.2 mM $CaCl_2$, 1.2 mM $MgCl_2$, 1.2 mM $KH_2PO_4$, 20 mM HEPES pH 7.4, 25 mM $NaHCO_3$, and 0.1% bovine serum albumin) supplemented with 2.8 mM glucose solution at 37 °C in 5% $CO_2$. Afterward, groups of ten islets that were size-matched between groups were transferred to a 96-well plate with KRBH solution containing low glucose (2.8 mM) or high glucose (16.8 mM). After incubation for 1 h, the supernatant was collected, and islets were lysed overnight in a 20% acid:80% ethanol solution. Insulin was then measured in supernatants and lysates using the Ultra Sensitive mouse insulin ELISA (Crystal chem). Insulin release was calculated as the percentage of total islet insulin content per hour.

**RNA-seq analysis**. RNA-seq was performed as previously described[31]. Briefly, RNA was isolated from livers of DIO mice treated with IXA4 or vehicle (n = 4 per condition) using the Zymo Research Quick-RNA Miniprep Kit according to the manufacturer's instructions. RNA sequencing was performed by BGI Americas on the BGI proprietary platform (DNBseq), providing single-end 50 bp reads at 20 million reads per sample. Alignment of the sequencing data to the *Mus musculus* MGSCv37 NCBI build 37.2 was performed using DNAstar Lasergene SeqManPro. Assembled data were then imported into ArrayStar 12.2 with QSeq (DNAStar Inc) to quantify gene expression and normalized reads per kilobase million (RPKM). Differential expression analysis and statistical significance calculations between conditions were assessed using DESeq in R using a standard negative binomial fit for the aligned counts data and are described relative to the indicated control (Supplementary Data 1). Gene Ontology (GO) enrichment analysis was performed using Panther (geneontology.org) on all transcripts which were significantly ($P_{adj}$ <0.05) downregulated/upregulated as observed by DESeq. Gene set enrichment analysis (GSEA) was performed using denoted genesets from GO on the GSEA platform for the mouse genome (http://www.informatics.jax.org/vocab/gene_ontology)[37,38]. Genesets used to prepare Fig. 1e were defined as previously described and shown in Supplementary Data 2[31]. Genes used to report on gluconeogenesis in Fig. 3c are shown in Supplementary Data 4.

**Real-time quantitative PCR (RT-qPCR) analysis**. RNA was extracted from cells or tissues using the Zymo Research Quick-RNA Miniprep Kit according to the manufacturer's instructions. RT-qPCR was performed on complementary DNA synthesized using the High-capacity Reverse Transcription Kit (Applied Biosystems catalog no. 4368814). cDNA was amplified using PowerSYBR Green Master

Mix (Applied Biosystems catalog no. 4367659) and primers purchased from IDT. Primer sequences used are listed below:

| Mouse gene | Forward primer | Reverse primer |
|---|---|---|
| Dnajb9/Erdj4 | CTTAGGTGTGCCAAAGTCTGC | GGCATCCGAGAGTGTTTCATA |
| Hspa5/Bip | GTCCAGGCTGGTGTCCTCTC | GATTATCGGAAGCCGTGGAG |
| Chop | GGAGCTGGAAGCCTGGTATG | TGTGCGTGTGACCTCTGTTG |
| Foxo1 | GTGAACACCATGCCTCACAC | CACAGTCCAAGCGCTCAATA |
| G6pc | TCACTTCTACTCTTGCTATCTTTCG | CCCAGAATCCCAACCACAAG |
| Pck1 | TTGAACTGACAGACTCGCCCT | TGCCCATCCGAGTCATGA |
| Dgat2 | CCGCAAAGGCTTTGTGAA | GGAATAAGTGGGAACCAGATCAG |
| Xbp1s | GAGTCCGCAGCAGGTG | GTGTCAGAGTCCATGGGA |
| Ins1 | GAAGCGTGGCATTGTGGAT | TGGGCCTTAGTTGCAGTAGTTCT |
| Ins2 | AGCCCTAAGTGATCCGCTACAA | CATGTTGAAACAATAACCTGGAAGA |
| Scd1 | CCTCCGGAAATGAACGACGAG | CAGGACGGATGTCTTCCA |
| Srebf1c | AACGTCACTTCCAGCTAGAC | CCACTAAGGTGCCTACAGACC |
| Ire1a | ACGAAGGCCTGACGAAACTT | ATCTGAACTTCGGCATGGGG |

**Immunoblotting.** Tissue pieces were homogenized using the Bullet Blender bead homogenizer (Braintree Scientific) and lysed in RIPA buffer (50 mM Tris, pH 7.5, 150 mM NaCl, 0.1% SDS, 1% Triton X-100, 0.5% sodium deoxycholate) containing protease and phosphatase inhibitor cocktail (Roche). Total protein concentration was quantified using the Pierce BCA assay kit. Lysates were denatured with 1× Laemmli buffer containing 100 mM DTT and boiled before being separated by SDS-PAGE. Proteins were transferred onto nitrocellulose membranes (Bio-Rad) for immunoblotting and blocked with 5% milk in Tris-buffered saline, 0.5% Tween-20 (TBST) prior to overnight incubation at 4 °C with primary antibodies. Membranes were washed in TBST, incubated with IR-Dye conjugated secondary antibodies, and analyzed using the Odyssey Infrared Imaging System (LI-COR Biosciences). Quantification was carried out using LI-COR Image Studio software.

**Histological analysis.** The liver and pancreas were processed for histology by the Histology core facility at the Scripps Research Institute. Samples were fixed in formalin, dehydrated, and embedded in paraffin. Tissues were cut into 5-μm-thick sections and subject to H&E staining. For Sirius red staining, samples were dewaxed and hydrated prior to incubation with Sirius red solution for 1 h at room temperature. Slides were then washed in acidified water (5 mL glacial acetic acid:1 L distilled water), dehydrated in ethanol, and cleared using xylene prior to mounting. Picro-Sirius red solution was purchased from Abcam (catalog no. ab246832). Quantification of the islet area from histological sections was performed using ImageJ software[39]. Hepatic steatosis was assessed by oil red-O staining in OCT-embedded cryosections using Oil Red O Stain Kit (Abcam catalog no. ab150678) following the manufacturer's instructions. Briefly, slides were placed in propylene glycol for 2 min and then incubated in oil red-O solution for 6 min at room temperature. A mixture of 85% propylene glycol in distilled water was prepared, and the tissue sections were incubated for 1 min at room temperature. The slides were rinsed twice with distilled water and then thoroughly rinsed in tap water at room temperature. The sections were then mounted in aqueous mounting media and scanned using a Leica AT2 slide scanner. Quantification of the stained area from histological sections was performed using ImageJ software[39].

**Quantification of liver lipids.** Liver lipids were extracted using the Folch method[40]. Briefly, 10 mg of the liver was weighed and homogenized in 50 mM Tris-HCl, 250 mM sucrose, and 1 mM EDTA pH 7.4, using the Bullet Blender bead homogenizer (Braintree Scientific) for 4 min. The homogenate was transferred to a glass tube and made up to 1 mL with PBS; 4 mL of a chloroform-methanol mixture (2:1) was then added and the mixture vortexed for 1 min. The glass tube was then spun at $700 \times g$ for 10 min in a centrifuge and the chloroform bottom phase transferred to a new glass tube using a 9-inch glass Pasteur pipette and dried under nitrogen. Lipids were then dissolved in 1 mL chloroform with 2% Triton X-100 and dried under nitrogen again. The dried samples were then re-dissolved in 1 mL of distilled water and incubated in a 37 °C water bath for 15 min with intermittent vortexing every 5 min. Levels of TG and total cholesterol in 20 μl of the sample were measured using commercial kits from Wako Diagnostics (LabAssay Trigly-ceride catalog no. 632-50991; Cholesterol E kit catalog no. 999-02601).

**Statistics.** All data is expressed as mean ± SEM for the indicated number of biological replicates. Statistical analysis was performed as indicated in the figure legends. Samples were excluded based on outlier testing, as appropriate, where indicated.

**Reporting summary.** Further information on research design is available in the Nature Research Reporting Summary linked to this article.

## Data availability

The raw data that support findings in this paper are available as Source Data included with this manuscript. RNA-seq data are available at the public National Center for Biotechnology Information Gene Expression Omnibus repository under the data identifier GSE162567. Source data are provided with this paper.

## Code availability

Code for standard open-source DESEQ differential gene expression RNA-seq analysis used in R statistical software is available from the corresponding authors upon reasonable request.

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

## Acknowledgements

This work was funded by NIH grants AG046495 and DK123038 to R.L.W., and DK114785 to E.S. We thank Jonathan Lin (Stanford) for the critical reading of the manuscript. We also thank Jeff Kelly and Kyunga Lee (Scripps) for helpful discussions, and Lars Plate (Vanderbilt) for experiments related to this work.

## Author contributions

A.M., B.P.K., B.R., E.S., and R.L.W. designed the research. A.M., B.P.K., B.R., J.M.D.G., A.A., A.G., and A.S. performed in vivo experiments. A.M., B.P.K., B.R., A.A., and V.A. performed in vitro experiments. A.M., J.M.D.G., E.T.P., and R.L.W. analyzed RNA-seq data. E.S. and R.L.W. supervised all aspects of this work. A.M., B.P.K., E.S., and R.L.W. wrote the manuscript with input from other co-authors.

## Competing interests

R.L.W. is an inventor on a patent describing IRE1/XBP1s-activating compounds, including IXA4, and is a scientific advisory board member and shareholder in Protego Biopharma, which has licensed these compounds for translational studies. The authors declare no other competing interests.
