## [Peer Review File · Nature Communications]

Reviewers' Comments:

Reviewer #1:

Remarks to the Author:

For this revised manuscript, can the authors provide some discussion or insight into the potential mechanistic basis (control) underlying the observation that IXA4 can activate/increase IRE1/XBP1s signaling, but has no effect on RIDD pathway? Because IRE1 RNase mediates the activation of both XBP1s and RIDD pathways, it is unusual that IXA4 treatment increases XBP1s signal without any effect on RIDD pathway.

Additionally, both AST and ALT levels (not only ALT levels), or ratios of AST/ALT, should be provided, as an indicator of liver injury.

Reviewer #2:

Remarks to the Author:

The manuscript of Madhavan et al. showed that a compound named IXA4 is able to selectively activate IRE1-XBP1s axis in liver and that treating diet-induced obese mice with IXA4 may improve systematic glucose metabolism, reduce fatty liver and promote insulin secretion following a glucose challenge. IXA4 is a putative activator of the IRE1a-XBP1s pathway. The findings are certainly of pharmaceutical potential in treating metabolic disorders, and the manuscript is well written. However, this revised study remains largely descriptive and in the opinion of this reviewer, remains premature and preliminary as many of my prior concern were not addressed experimentally.

1. The authors added one experiment to show XBP1s protein level and Xbp1 mRNA splicing in the liver of mice treated for 8 w in Fig 1g. However, it lacks the analysis of XBP1s protein levels and IRE1a activation in vivo at different time points in target tissues such as the liver and pancreatic islets, as Rev 1 and I requested previously. I am indeed questioning whether the drug hit the pancreatic β cells? Gene expression of 1-2 targets is not sufficient in my opinion. Additional analyses at the mRNA/protein levels of ER homeostasis markers, such as XBP1 splicing, and phosphorylation and total of IRE1a proteins are necessary to demonstrate the acute and chronic role of IXA4 on IRE1a. Without the latter (p-IRE1a), it remains unclear whether IXA4 directly activates IRE1a in vivo.

2. In the revised version, the authors introduced additional drug treatment with an IRE1a inhibitor to counter the effect of IXA4, rather than the genetic approach requested by both Rev 1 and myself. As both inhibitors are unproven IRE1a inhibitors in vivo, these new experiments brought more questions than answers in my opinion. I think the comment from the last round should be addressed: "Most importantly, the use of drugs with broad specificity (such as 4 μ 8C) in vivo and in vitro, without the inclusion of genetic KO models (such as IRE1a and XBP1 KO models), can be fraught with artifacts and non-specific effects of these drugs." It does not have to be mouse models, in vitro KO hepatocytes and β cells will be sufficient.

3. I agree with the Rev 1 that energy expenditure of the cohorts subjected to drug treatment is important. I am concerned about the off-target or more profound effect of the drug in the brain or overall systemic energy metabolism. The energy expenditure will provide some insights into this.

4. Equally importantly, the lack of mechanistic insights in the beneficial effect of IXA4 in vivo render this study incremental from their 2020 paper. Whether and how does IXA4 affect β cell function (or possibly β cell identity) and insulin biogenesis remained unexplored. The only evidence for the effect of IXA4 on proinsulin maturation and process is Fig. 4d proinsulin:insulin ratio. As the effect is mild and as there is no other experimental data to support this finding, I am not convinced that's what the drug does in β cells. To me, although this study is the use of a new drug in an in vivo study, a more important and deeper question is how it works in vivo, and whether it does so by specifically targeting IRE1a. As this manuscript does not address these questions, I can not support it for publication in Nat Communication.

The same Xbp1 mRNA splicing data were used twice in Fig S1a and S3a.

Reviewer #3:

Remarks to the Author:

In this manuscript, Madhavan et al. show that pharmacologic activation of the UPR sensor IRE1/XBP1 via IXA4 treatment can improve glucose metabolism, insulin action in the liver and enhance pancreatic beta cell function and insulin homeostasis. Furthermore, the authors present evidence that the activation of IRE1/XBP1 signaling can protect liver from steatosis in obese mice. This is an interesting study with potential therapeutic implications and will be of significance for the field. While there are multiple IRE1/XBP1 inhibitors this study presents the first activator and its effects on metabolic syndrome. Data and the interpretation of results are sound. Therefore, as presented the manuscript will advance the field.

RESPONSE TO REVIEWERS – NCOMMS-21-34487

REVIEWER 1

“For this revised manuscript, can the authors provide some discussion or insight into the potential mechanistic basis (control) underlying the observation that IXA4 can activate/increase IRE1/XBP1s signaling, but has no effect on RIDD pathway? Because IRE1 RNase mediates the activation of both XBP1s and RIDD pathways, it is unusual that IXA4 treatment increases XBP1s signal without any effect on RIDD pathway.”

Response: We have included expanded discussion to explain why our compound IXA4 selectively activates IRE1/XBP1s splicing relative to induction of IRE1-mediated RIDD. We have previously shown (Grandjean et al., 2020 *Nat Chem Biol* 16:1052) that IXA4 induces transient, moderate activation of IRE1 involving modest increases in IRE1 autophosphorylation and XBP1s splicing (~40% of that observed with global ER stressors). This mild level of activation is not sufficient to bring about RIDD activity, which is only induced upon chronic, high levels of IRE1 activation. In fact, in work performed with Peter Walter’s group, we have found that IXA4 does not stimulate formation of the high molecular weight clusters of IRE1 that are associated with RIDD activity in cell culture models.

“Additionally, both AST and ALT levels (not only ALT levels), or ratios of AST/ALT, should be provided, as an indicator of liver injury.”

Response: We now include both AST and ALT levels in the revised manuscript (see **Fig. S1v,w**). These two markers of liver health are not altered in IXA4-treated DIO animals, confirming that IXA4 treatment does not induce liver damage.

REVIEWER 2

1. *“The authors added one experiment to show XBP1s protein level and Xbp1 mRNA splicing in the liver of mice treated for 8 w in Fig 1g. However, it lacks the analysis of XBP1s protein levels and IRE1 α activation in vivo at different time points in target tissues such as the liver and pancreatic islets, as Rev 1 and I requested previously. I am indeed questioning whether the drug hit the pancreatic β cells? Gene expression of 1-2 targets is not sufficient in my opinion. Additional analyses at the mRNA/protein levels of ER homeostasis markers, such as XBP1 splicing, and phosphorylation and total of IRE1 α proteins are necessary to demonstrate the acute and chronic role of IXA4 on IRE1 α . Without the latter (p-IRE1 α), it remains unclear whether IXA4 directly activates IRE1 α in vivo.”*

Response: The Reviewer is mistaken. Reviewer 1 made no such request in prior comments, and this Reviewer asked only that we provide *“Other analyses at the mRNA/protein levels, such as XBP1 splicing, phosphorylation of IRE1 α , PERK, and p-eIF2 α are necessary to demonstrate the acute and chronic role of IXA4 on UPR”*. In response, in the revised manuscript, we added data showing that IXA4 treatment increased *Xbp1s* mRNA levels in the liver of chow-fed mice measured 4 h after dosing (**Fig. 1c**) which, as expected, corresponded with higher expression of the canonical IRE1/XBP1s target gene *Dnajb9/Erdj4* (**Fig. 1b,c**). Moreover, we showed that co-treatment with the specific IRE1 inhibitor STF-083010 blocked this IXA4-stimulated induction of *Xbp1s* and *Dnajb9/Erdj4* expression (**Fig. 1c**). In addition, we added data showing that XBP1s

protein levels are increased in the liver of DIO mice treated with IXA4 for 8 weeks (**Fig. 1g,h**). Thus, we offered clear evidence that IXA4 activates IRE1 and stimulates XBP1s splicing in an acute (4 h) and chronic fashion in treated mice. For the Reviewer to now request greater “*analysis of XBP1s protein levels and IRE1a activation in vivo at different time points in target tissues such as the liver and pancreatic islets*” is unreasonable. The new data added in the revision, together with the extensive RNAseq profiling included (well beyond “*expression of 1-2 targets*”; e.g., **Fig. 1e,f, S1h**), demonstrates conclusively that IXA4 is a selective activator of IRE1/XBP1s signaling in mice. The additional data now requested will add little to the comprehensive profiling of IRE1/XBP1s activity we already provide.

We have published that IXA4 modestly increases IRE1 phosphorylation in cell culture and that IXA4-induced IRE1/XBP1s activation requires IRE1 autophosphorylation (Grandjean et al., 2020 *Nat Chem Biol* 16:1052). Detecting modest changes in IRE1 phosphorylation in tissues, as the Reviewer requests, is not technically feasible with the antibodies currently available. These tools can only detect large changes in IRE1 phosphorylation, such as those induced by global ER stressors. The more moderate, transient changes induced by our compounds cannot be effectively detected in tissues in this way. A better approach to monitor the status of IRE1 phosphorylation, and frankly one that is more physiologically relevant, is to gage the functional output of IRE1 phosphorylation by evaluating the induction of its downstream target genes, as we have done here using qPCR and RNAseq. The extensive transcriptional profiling in cells and tissues we present clearly indicates that IXA4 selectively activates IRE1/XBP1s signaling, as reflected in increased expression of numerous of its established target genes.

The Reviewer is likely aware of the technical challenges in extracting intact mRNA from whole pancreas. We tried to isolate mRNA from pancreas and faced these widely reported problems (e.g., RNA degradation by RNAses). However, and directly addressing the issue of whether “*the drug hit the pancreatic β cells?*”, we show that pancreatic islets isolated from IXA4-treated DIO mice incubated overnight in IXA4-free media secrete more insulin in response to glucose (**Fig. 4**). This improvement in GSIS must be a reflection of the IXA4-induced proteome remodeling that the compound stimulated while the DIO mice were being treated with it, a beneficial reprogramming that persists for 24 h in the absence of compound. This data clearly shows that IXA4 does act on pancreatic beta cells in treated mice.

Our work provides *the most comprehensive profiling* of any UPR-modulating compound described to date in a disease model. We used qPCR, protein level changes, and transcriptional profiling to clearly demonstrate that IXA4 selectively activates IRE1/XBP1s signaling in vivo. It is exactly *for this reason* that our unique and well-characterized compounds have been sought by >20 labs studying the UPR both in vitro and in vivo. The Reviewer errs when suggesting that our compounds lack selectivity, or that this is not sufficiently established in our revised manuscript.

2. “*In the revised version, the authors introduced additional drug treatment with an IRE1a inhibitor to counter the effect of IXA4, rather than the genetic approach requested by both Rev 1 and myself. As both inhibitors are unproven IRE1a inhibitors in vivo, these new experiments brought more questions than answers in my opinion. I think the comment from the last round should be addressed: “Most importantly, the use of drugs with broad specificity (such as 4 μ 8C) in vivo and in vitro, without the inclusion of genetic KO models (such as IRE1a and XBP1 KO models), can be fraught with artifacts and non-specific effects of these drugs.” It does not have to be mouse models, in vitro KO hepatocytes and β cells will be sufficient.*”

Response: The assertion that 4 μ 8c and STF-083010 “*are unproven IRE1 α inhibitors in vivo*” is incorrect. These compounds have been and continue to be widely used to define the importance of IRE1 signaling in vivo in multiple disease models (e.g., PMID: 28137856, 30728287, 21081713, 29394934, 29457838). These IRE1 inhibitors are acknowledged in the field as being some of the best pharmacological tools available to selectively inhibit IRE1 activity in cells and mice in the precise manner we use them in our work. As we described more extensively in the rebuttal, *genetic approaches to block IRE1/XBP1s activity in vivo are not appropriate for this study*. In both, cells and mice, genetic depletion of this stress pathway elicits major changes to cellular and tissue physiology, alterations that invariably activate compensation mechanisms. While some may think that this is selective, *these types of manipulations are consistently associated with ‘off-target’ altered signaling through other pathways that compensate and thus confound interpretation of findings*. Pharmacologic approaches can also elicit off-target effects, but the ability to control treatment time and the use of two structurally-distinct IRE1 inhibitors (i.e., 4 μ 8c and STF-083010) considerably minimizes off-target concerns in our studies. Our data show that the improvements in glucose homeostasis and liver steatosis seen in IXA4-treated DIO mice are dependent on IRE1 activity, for co-treatment with an IRE1 inhibitor eliminates these beneficial effects. We also demonstrate that in primary hepatocytes these changes are dependent on IRE1 activity using two distinct IRE1 inhibitors, an established approach in the field to define the need for IRE1. Furthermore, we also used both IRE1 inhibitors and genetic *Ire1* knockdown to establish that IXA4 regulates insulin secretion in Min6 cells in a IRE1-dependent manner. Our findings are clear: the effects of IXA4 in vitro and in vivo require IRE1 activity.

We also note that we have published that IXA4 does *not* activate IRE1/XBP1s signaling in *Ire1*-deficient MEFs (Grandjean et al., 2020 *Nat Chem Biol* 16:1052). We highlight this finding in the revised manuscript on page 4.

3. “*I agree with the Rev 1 that energy expenditure of the cohorts subjected to drug treatment is important. I am concerned about the off-target or more profound effect of the drug in the brain or overall systemic energy metabolism. The energy expenditure will provide some insights into this.*”

Response: Respectfully, we see little value in this request. There is no difference in body weight or food intake in DIO mice treated with IXA4. Thus, there is no need to measure energy balance parameters in these mice. Our response was sufficient for Reviewer 1 who originally brought up this point. We cannot discern what Reviewer 2 thinks these experiments will add to the story; treated mice do not eat or weigh less, which indicates that the brain is not likely to play a significant role. Furthermore, we have found that IXA4 does not cross the blood-brain barrier and does not activate IRE1/XBP1s signaling in the brain. Thus, not only is it unlikely that IXA4 alters energy expenditure, but it is also impossible that it may do so by acting on the brain. We will be happy to add this data showing the lack of an effect of IXA4 treatment on brain IRE1/XBP1s signaling.

4. “*Equally importantly, the lack of mechanistic insights in the beneficial effect of IXA4 in vivo render this study incremental from their 2020 paper. Whether and how does IXA4 affect β cell function (or possibly β cell identity) and insulin biogenesis remained unexplored. The only evidence for the effect of IXA4 on proinsulin maturation and process is Fig. 4d proinsulin:insulin ratio. As the effect is mild and as there is no other experimental data to support this finding, I am not convinced that’s what the drug does in β cells. To me, although this study is the use of a new drug in an in vivo study, a more important and deeper question is how it works in vivo, and whether it*

does do by specifically targeting IRE1a. As this manuscript does not address these questions, I can not support it for publication in Nat Communication.”

Response: The Reviewer is mistaken. *Our 2020 paper has no data on the effects of IXA4 on beta cells or insulin secretion.* None. All of these are new findings that constitute the first report that pharmacological activation of IRE1/XBP1s can have beneficial effects in beta cells and pancreatic islets in the context of obesity. In the revised manuscript we show that chronic IXA4 treatment did not alter pancreas morphology or islet number (**Fig. 4a**), indicating that IXA4 had no major impact on beta cell proliferation. We also show that IXA4 treatment did not alter islet insulin content (**Fig. 4b**) or expression of *Ins1* or *Ins2* (**Fig. S4e**), showing that IXA4 does not affect insulin expression or biogenesis. The primary effect of IXA4 treatment is a reduction in the plasma proinsulin:insulin ratio following an oral glucose bolus in DIO mice (**Fig. 4d**). This finding is consistent with IXA4 acting in beta cells to promote processing of proinsulin to mature, active insulin, a fitting role for a modulator of the UPR response. We agree with the Reviewer that greater description of the molecular basis of this effect is a worthwhile effort, but we feel that it is well beyond the scope of the present study. Multiple studies using genetic tools have indicated that chronic IRE1 hyperactivity leads to beta cell death and a reduction in islet insulin content. In contrast, our work demonstrates that a mild and selective pharmacological increase in IRE1/XBP1 activity can enhance glucose-stimulated insulin secretion. This is the first report that increased IRE1/XBP1s signaling can enhance beta cell function in obesity, not exactly an incremental finding.

The Reviewer does not seem to appreciate that our work shows, for the first time, the ability to pharmacologically and selectively activate IRE1/XBP1s signaling in vivo, and that our study *as is* outlines the potential of this approach to correct metabolic dysfunction in obese-insulin resistant mice by stimulating adaptive remodeling in multiple tissues. This is a *major finding* in the field and one that offers a new direction to probe the therapeutic potential of IRE1/XBP1s activity in the context of diverse diseases. Underscoring the importance of this work, since we posted the initial version of this manuscript in *bioRxiv*, we have received requests for IXA4 from more than 20 labs eager to use this heretofore unprecedented pharmacological tool to examine the importance of IRE1/XBP1s signaling in numerous disease models in which ER stress is a player. Our work is a milestone in the development of pharmacologic approaches to modulate UPR signaling in vivo, and it provides a rationale for developing selective UPR modulators for translational studies.

5. *“The same Xbp1 mRNA splicing data were used twice in Fig S1a and S3a.”*

Response: The Reviewer is correct. We apologize and have removed the *Xbp1* mRNA splicing data from **Fig. S3a**.

REVIEWER 3

This Reviewer had no further comments and deemed the revised work suitable for publication.